**Data Availability Statement:** We do not have explicit consent from our participants to make our data publicly available. In fact, because of the

# Traumatic experiences and place of occurrence: An analysis of sex differences among a sample of recently arrived immigrant adults from Latin America

**Laura X. Vargas**[1]*, **Mary D. Sammel**[1,2], **Therese S. Richmond**[3], **Connie M. Ulrich**[3], **Zachary D. Giano**[4], **Lily Berkowitz**[1], **C. Neill Epperson**[1]

1 Department of Psychiatry, School of Medicine, University of Colorado-Anschutz Medical Campus, Aurora, CO, United States of America, 2 Department of Biostatistics and Informatics, Colorado School of Public Health, University of Colorado-Anschutz Medical Campus, Aurora, CO, United States of America, 3 Department of Behavioral Health Sciences, University of Pennsylvania School of Nursing, Philadelphia, PA, United States of America, 4 Center for Innovative Design & Analysis, Colorado School of Public Health, University of Colorado Anschutz Medical Campus, Aurora, CO, United States of America

* laura.x.vargas@cuanschutz.edu

## Abstract

With increasing violence, political, and economic instability in Latin America, there is a record number of migrants crossing the U.S. southern border. Latin American migrants are often exposed to traumatic events before leaving their home country and during migration. While prior studies document that sex may play a role in types of traumatic exposure, few studies compare differences in traumatic exposure by sex and place of occurrence of recently arrived immigrants. Addressing this gap, we recruited 120 adults who had recently crossed the U.S.-Mexico border. Participants completed questionnaires to characterize trauma exposures in their home country and during their migration journey. Results found that men reported higher levels of exposure to combat situations, while women were more likely to experience sexual assault. Both combat exposure and sexual traumas occurred more often in home countries than during migration. More than half of the full sample reported being threatened with a firearm. These data confirm gender differences in type of trauma and that exposures in the country of origin may provide the impetus to migrate.

## Introduction

Historically, migration from Latin America to the U.S. predominantly consisted of single, working-age males seeking economic opportunity [1]. Latin American migrants from four countries (i.e., El Salvador, Guatemala, Honduras, and Mexico) account for over 90% of all apprehensions by United States (U.S.) Customs and Border Patrol (CBP) at the southern border, but the proportion of migrants from other origin countries in Latin America has increased in recent years. Rising violence in the region contributes to changes in the demographic composition of migrants. Given the demographic transition in the Latinx migrant population to the U.S. and the rise in forced migration from Latin America, it is not surprising that limited

sensitive nature of this data and the vulnerability of this population, we have told our participants that this data is only available in a de-identified way to members of the research team. We realize that we are in an era of open data, but we also know of the reluctance of certain understudied and extremely important populations to participate in research. Part of our ability to collect this data is the trust that participants place in us. We value their trust greatly and it is vital to respect our commitment to the trust placed in us. However, if specific data requests deriving from the publication of this article should arise, those requests can be directed to the Psychiatry Research Innovations office. The Psychiatry Research Innovations (PRI) is a centralized research support infrastructure with a specific mission to advance academic success of faculty members in the Department of Psychiatry, at the University of Colorado School of Medicine. Within this role, the PRI support various research activities through five service cores: i) sponsored program management service (pre/post award, locating funding opportunities, etc); ii) research operations and administration (regulatory submission, database set up and management, project management, etc); iii) biostatistics (creating study design and analytical plan, data analysis, etc); iv) clinical support focusing on quality improvement and program evaluation type of projects; and v) research education and training (didactics and presentations to build research skills, mentoring programs, etc). Faculty will be provided support in any of these areas on as needed basis. Specific queries can be directed towards the PRI Director, Dr. Merlin Ariefdjohan: MERLIN.ARIEFDJOHAN@CUANSCHUTZ.EDU.

**Funding:** Dr. Laura X. Vargas received funding for this study from a National Institute on Minority Health and Health Disparities (NIMHD) K01 Career Development Award (1K01MD015768, PI: Vargas) https://www.nimhd.nih.gov/ The funders had no role in study design, data collection and analysis, decision to publish, or preparation of the manuscript.

**Competing interests:** The authors have declared that no competing interests exist.

prior research has examined sex differences in trauma exposure among recently arrived Latinx immigrants. These changes in demographic patterns offer the opportunity to study immigrants from a wider array of countries of origin. The purpose of this research is to fill a gap in understanding how traumatic experiences of recently arrived immigrant adults from Latin America may vary by sex and by place of occurrence (i.e., in the home country, during the migration journey, or both) prior to their arrival to the U.S.

The U.S. Latinx population has increased nearly nine-fold since 1960 and is projected to grow to 107 million by 2065 [2]. Exposure to violence in Latin America is a significant contributor to migration of adults and families to the U.S. [3, 4]. Latin America accounts for 36% of the world's homicides despite representing only 8% of the world's population [5]. Migration as a result of forced displacement is rising [6] and contributes to the increasing diversity of migrants (i.e. more women, children, older adults). Several studies in various Latin American countries report a wide exposure to potentially traumatic events [7–13].

Many studies of traumatic exposure conducted in Latin American countries either report no overall association between prior traumatic exposure (PTE) and sex or that males are more likely to report a PTE [7, 9–11, 14, 15]. These studies also suggests that sex differences are dependent on the type of event; for example, most studies find that females are significantly more likely to report experiencing sexual violence [7, 9–11, 15]. Thus, as violence is a widespread problem throughout the Latin American region, it is important to examine trauma exposure differences by sex among newly arrived immigrants from Latin America.

While these studies illustrate that people migrating from Latin America may experience potentially traumatic events in their home countries, they do not specifically refer to the experiences of migrants. Studies highlight that the relationship of migration and mental health [16] is mediated by trauma exposure, and various other covariates including sex [12, 17–19]. Very few studies focus on in depth research of specific trauma experiences of women immigrants [19]. Some studies have focused on the impacts of exposure to trauma and mental health among Latinx immigrants [11, 20–22], though few have studied exposure to trauma among recent Latinx immigrants [23, 24]. The rationale for our study is that we present a first time comparison of traumatic experiences of recent immigrant adults from Latin America by sex and place of occurrence.

## Methods

### Design and setting

This cross-sectional descriptive study was reviewed and approved by the Colorado Multiple Institutional Review Board (COMIRB), the Institutional Review Board (IRB) for the University of Colorado Anschutz Medical Campus. The setting for this study was a humanitarian respite center located on the U.S. side of the U.S.-Mexico border.

### Participants

Adults ages 18+, Spanish-speaking, who arrived in the U.S. within the past 14 days of the date of interview, were from a Latin American country, and able to give informed consent were recruited to participate in the study. In the rare case that potential participants only spoke English were eligible for participation; potential participants who only spoke a non-Spanish indigenous or regional dialect and did not understand or speak Spanish or English were excluded.

### Data collection

We approached 147 individuals for recruitment after having their basic needs met and 120 were enrolled and surveyed if they had at least two hours to wait for transportation at a

humanitarian respite center in McAllen, Texas. Study recruitment took place from April 2022 through March 2023. The study purpose and procedures were explained, all questions answered, and verbal informed consent obtained. Participants were interviewed in a private area to ensure confidentiality. To thank participants for their time, we provided a $50 gift card at survey completion to a major retail store in the U.S. Data were stored according to a unique code given to each participant in lieu of participant names. Some of the reasons provided by those approached for recruitment but who did not enroll in the study were: they did not have sufficient time, they were tired from their journey, or they preferred not to take part in the study.

## Measures

Exposure to trauma. Trauma exposure is defined as exposure to violence, crime, sexual exploitation, deprivation, discrimination, or other intentional and unintentional traumatic experiences that occur before and during migration to the U.S. A combination of measures were used to capture a history of potentially traumatic experiences: (1) an adapted version in Spanish of the validated Harvard Trauma Questionnaire (HTQ) [25–27] Peruvian version that measures (a) material deprivation, (b) war-like conditions, (c) bodily injury, (d) forced confinement/coercion, (e) disappearance/death/injury of loved ones, (f) witnessing violence to others, and (g) kidnapping and extortion; (2) exposure to discrimination from the National Latino and Asian American Study (NLAAS) included items assessing everyday experiences of discrimination and perceived discrimination that show high internal consistency both in Spanish and English [28]; and (3) a multiple choice question asking immigrants about their reasons for migrating, with the following answers: settlement (long term/permanent stay); employment; education or training; marriage, family reunification, or family formation; forced displacement (refugees, asylum seekers, temporary protection, etc.); displacement due to climate events; or other reason: where participants were asked to elaborate on that reason.

Demographic information. Individual demographic characteristics from the World Health Organization World Mental Health Composite International Diagnostic Interview (WHO WMH-CIDI) [29] include sex (male/female), age, marital status, education, main language spoken at home, and employment in the previous year (i.e. the question asks "how many months did you work in previous 12 months?"). We complemented CIDI demographics with questions about country of origin, the migration journey, such as time since leaving home country, prior migration experience, illnesses, and access to food, water, and medical care. We also ask two questions to understand participants English language abilities, asking them to self-rate (on a scale of 1–10 with 1 being "not at all" and 10 being "perfectly") their ability to understand spoken English and speak English, respectively.

## Statistical analysis

Frequency and descriptive statistics were generated with significance testing between males and females using Pearson Chi-square and t-tests where appropriate. Frequencies were calculated for the 45 trauma items for the full sample, males, and females; including the location of the trauma, when applicable. Two sets of testing were then conducted: the first evaluated the association between sex (male/female) and trauma type (yes/no) using Chi-square tests. A second Chi-square test was conducted to assess the association between sex (male/female) and a 4-level variable regarding the location/timing of trauma (no exposure/home country/during migration/both home country and migration). Due to small cell sizes, a number of comparisons used Fisher's exact test (noted in Table 3). Finally, we compared cumulative count of the

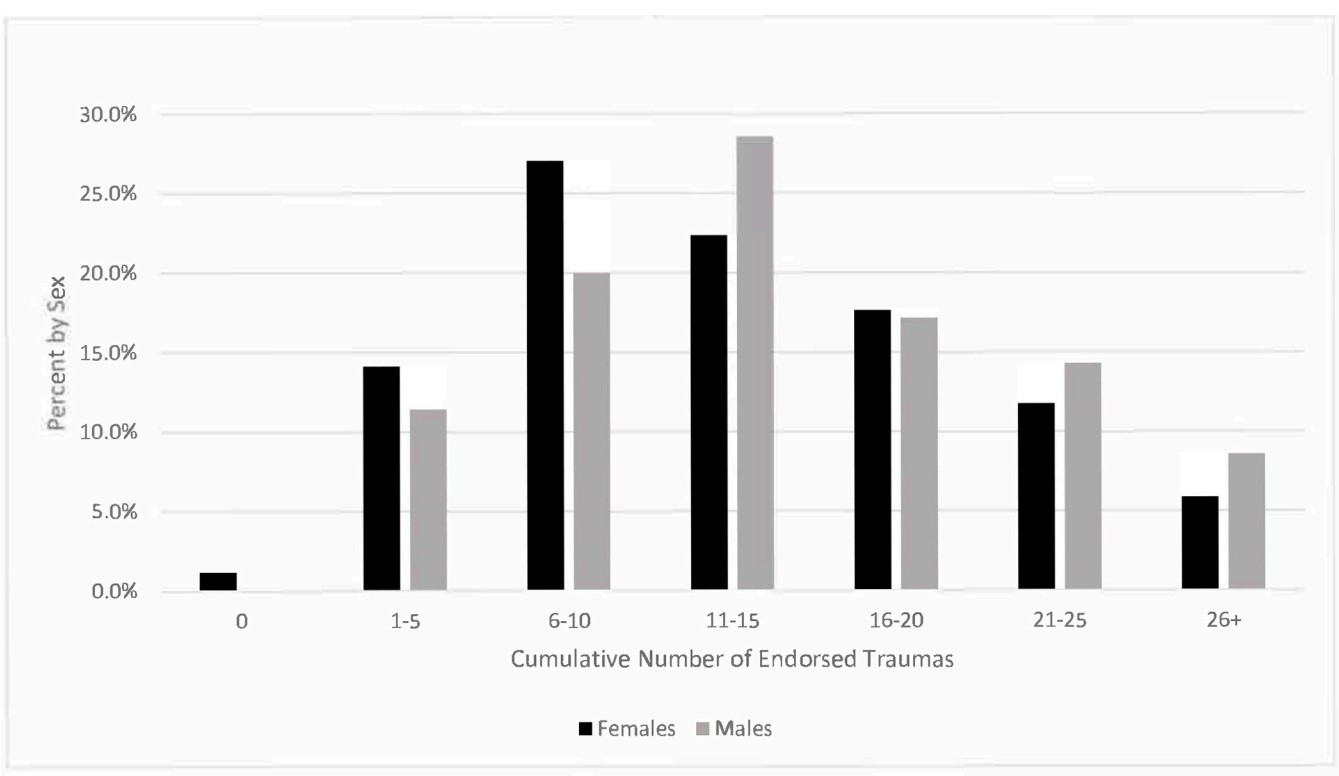

**Fig 1. Cumulative traumas by sex.**

number of trauma experiences by sex and graphed them by different occurrence categories (Fig 1. Cumulative Traumas by Sex).

Migration journey duration was calculated in days by subtracting departure date from interview date. The 15th day of the month was assigned to those who only recalled month and year of departure if departure took place prior to 1/1/2020. Three participants were missing the day of departure for a date after 1/1/2020 or had an otherwise unusable date (e.g., missing month or year), their journey length was considered missing and was excluded only from journey length analysis.

## Results

### Participant characteristics

The convenience sample consisted mostly of 120 asylum-seeking adults who were in transit to a U.S. destination. Although undocumented migrants may come to the U.S., in the current environment and with existing policies, most people at the study enrollment site are asylum-seeking immigrants.

Table 1 presents demographic and migration journey characteristics. Nearly 70% of the sample were female participants with an average age of 33.9 years. Male participants were significantly older (36.5) than female participants (32.8) (p = 0.020). Nearly 90% of respondents were traveling with children and the average number of children per respondent was 2.2. Male participants worked on average nearly two months more than female participants in the past year (p = 0.041). While Spanish was the predominant language spoken at home among participants, males self-rated their ability to speak and understand English as higher compared to

**Table 1. Sample demographics (N = 120).**

| | | | Range, if Applicable | Full Sample | Females | Males | p-value | |
|---|---|---|---|---|---|---|---|---|
| | | | | %/M (StDev) | %/M | %/M | | |
| *Demographics* | | | | | | | | |
| | Female | | | 69.7 | - | - | | |
| | Married | | | 39.3 | 32.9 | 55.6 | | |
| | Live w/Significant Other | | | 67.2 | 58.8 | 88.9 | | |
| | Pregnant/Spouse is Pregnant | | | 15.6 | 18.8 | 8.3 | | |
| | Age | | | *33.9 (8.2)* | *32.8 (7.9)* | *36.5 (8.5)* | **.020** | a |
| | Number of Children | | | *2.17 (1.3)* | *2.08 (1.2)* | *2.3 (1.4)* | .275 | a |
| | Months Employed in Past Year | | | *6.91 (4.9)* | *6.3 (5.1)* | *8.31 (4.1)* | **.041** | a |
| | Years of Education | | | | | | .141 | a |
| | | 0 | | 2.5 | 1.2 | 2.0 | | |
| | | 1–6 | | 27.4 | 27.1 | 27.8 | | |
| | | 7–9 | | 20.6 | 21.2 | 19.5 | | |
| | | 10–12 | | 24.8 | 22.4 | 30.6 | | |
| | | 13–15 | | 14.1 | 14.1 | 13.9 | | |
| | | 16+ | | 10.6 | 14 | 2.8 | | |
| *Language Abilities* | | | | | | | | |
| | Spanish Spoken at Home | | | 93.4 | 95.3 | 91.7 | .423 | b |
| | Ability to Understand English | | 1–10 | *3.2 (2.3)* | *2.8 (2.1)* | *3.8 (2.7)* | **.023** | a |
| | Ability to Speak English | | 1–10 | *2.6 (2.1)* | *2.3 (1.6)* | *3.3 (2.9)* | **.013** | a |
| *Immigration Characteristics* | | | | | | | | |
| | First time in U.S. | | | 90.9 | 92.9 | 86.1 | .236 | b |
| | First time Crossing U.S. Border | | | 80.2 | 82.4 | 75.0 | .358 | b |
| | Traveling with Children | | | 89.3 | 87.1 | 94.4 | .234 | b |
| | No food for more than a day | | | 72.7 | 72.9 | 72.2 | .936 | b |
| | Dehydration | | | 62.8 | 58.8 | 72.2 | .166 | b |
| | Illness/Injury | | | 58.7 | 60.0 | 55.6 | .653 | b |
| | Lack of Medical Care | | | 68.6 | 68.2 | 69.4 | .897 | b |
| | Mental Health Crisis | | | 75.2 | 77.6 | 69.4 | .344 | b |
| | Lack of Personal Hygiene | | | 65.3 | 65.9 | 63.9 | .835 | b |
| | Lack of Food for Children | | | 63.3 | 61.2 | 68.6 | .449 | b |
| | Apprehended Upon Arrival | | | 95.0 | 96.5 | 91.7 | .270 | b |
| | Days Apprehended (if applicable) | | | 2.0 | 1.9 | 2.1 | .105 | a |
| | Days from start of journey to interview | | | 131.2 | 104.1 | 200.1 | | |
| *Country/Region of Origin* | | | | | | | | |
| | Mexico | | | 1.7 | 2.4 | - | | |
| | Central America | | | 80.6 | 81.0 | 80.5 | | |
| | Caribbean | | | 6.8 | 5.9 | 8.3 | | |
| | South America | | | 10.9 | 10.7 | 13.2 | | |

Note: a = t-test, b = Chi Square test

females. Nearly 90% of the sample responded that this was their first time in the U.S., but men were more likely than women to endorse having been to the U.S. previously. Just over 80% was from Central America, with 10% from South America and nearly 7% were from the Caribbean. Additionally, 15.6% of the total sample were pregnant women or men who were travelling with a pregnant partner, and 94.4% of the sample were traveling with children.

**Table 2. Country demographics.**

| | | Full Sample | Females | Males | Migration Days | Migration Days | % Prolonged Stay in | % Held in |
|---|---|---|---|---|---|---|---|---|
| | | (%) | (%) | (%) | M/StDev | Median | Mexico | Mexico |
| *Country of Origin* | | | | | | | | |
| | Colombia | 1.7 | 2.4 | - | 9.0 (7.1) | 9.0 | 50.0 | 0.0 |
| | Cuba | 2.6 | 3.5 | - | 23.0 (4.0) | 23.0 | 100.0 | 33.3 |
| | Ecuador | 2.6 | 2.4 | 2.8 | 13.0 (6.9) | 9.0 | 33.3 | 0.0 |
| | El Salvador | 11.5 | 12.7 | 8.3 | 38.4 (74.7) | 15.0 | 100.0 | 50.0 |
| | Guatemala | 9.1 | 11.8 | 2.8 | 40.4 (94.7) | 12.0 | 90.9 | 20.0 |
| | Haiti | 4.2 | 2.4 | 8.3 | 1795.0 (1030.8) | 2042.0 | 100.0 | 0.0 |
| | Honduras | 21.4 | 21.2 | 22.2 | 98.6 (206.8) | 25.0 | 100.0 | 50.0 |
| | Mexico | 1.7 | 2.4 | - | 215.0 (14.1) | 215.0 | 100.0 | 100.0 |
| | Nicaragua | 38.6 | 35.3 | 47.2 | 25.9 (15.9) | 20.5 | 87.0 | 28.2 |
| | Peru | 4.9 | 3.5 | 8.4 | 7.6 (2.8) | 9.0 | 50.0 | 66.7 |
| | Venezuela | 1.7 | 2.4 | - | 50.0 (8.4) | 50.0 | 100.0 | 100.0 |

Top Reasons for Immigration

Forced displacement (refugees, asylum seekers, temporary protection, etc.); n = 63

Employment; n = 14

Other: Economic and political situation in home country; n = 13

All but 5% were apprehended upon arrival at the U.S. southern border and endorsed average time in apprehension of just over 2 days. Conditions of the migration journey showed that just over 72% experienced a lack of food for more than a day and over 62% experienced a lack of water for more than 24 hours. Nearly 60% experienced an illness or injury during their journey and 68% endorsed a lack health care services when they needed them. Sixty three percent of participants did not have access to food for their children for more than a day and 65% did not have access to adequate personal hygiene conditions and products. Nearly three quarters of participants endorsed experiencing at least one mental health problem during their journey. The average time from the date participants left their home country to the time of interview was 131 days.

## Country demographics and experiences of trauma

Table 2 presents country demographics. Participants came from eleven countries in Latin America and the Caribbean, with the largest portion of the sample from Nicaragua (38.6%). The number of days participants spent migrating (from the date they left their home country, to the date they were interviewed after arriving to the U.S.) ranged widely by country of origin with participants from Haiti migrating on average for over 2,000 days and some participants from countries like Peru and Colombia migrating in under 10 days on average. Most participants from countries other than Peru, Colombia and Ecuador spent a prolonged period of more than one week in Mexico, though a smaller proportion of those participants endorsed having been held or spent time in Mexico by force.

Table 3 presents descriptive statistics of experiences of trauma in the sample measured by the HTQ and four additional trauma exposure questions. The first column of results describes the percent of the sample that experienced each of the specific traumas. Results are ranked by percent of occurrence from high (top of the table) to low. The subsequent columns present the percent of males and females that did not experience each trauma, and if exposed to trauma, at what point in the trajectory it occurred. The final two columns present the chi-square

**Table 3. Harvard Trauma Questionnaire results.**

| | % of Trauma by Full Sample | Males | | | | Females | | | | $\chi^2$ (sex & 2-level trauma) | $\chi^2$ (sex & 4-level place) |
|---|---|---|---|---|---|---|---|---|---|---|---|
| | | No Exposure | Home Country | After | Both | No Exposure | Home Country | After | Both | | |
| Forced to hide | 75.0 | 19.4 | 22.2 | 38.9 | 19.4 | 28.2 | 9.4 | 40.0 | 22.4 | .148 | .261 |
| Lack of food or water | 70.0 | 36.1 | 5.6 | 25.0 | 33.3 | 28.2 | 9.4 | 28.2 | 34.1 | .327 | .784 |
| Confined to home because of danger outside | 70.0 | 27.8 | 0.0 | 55.6 | 16.7 | 31.8 | 44.7 | 4.7 | 18.8 | .335 | .473 |
| Extortion or robbery | 62.5 | 33.3 | 13.9 | 25.0 | 27.8 | 40.0 | 15.3 | 25.9 | 18.8 | .252 | .734 |
| Ill health without access to medical care | 59.2 | 41.7 | 13.9 | 11.1 | 33.3 | 41.2 | 12.9 | 18.8 | 27.1 | .536 | .736 |
| Forced evacuation under dangerous conditions | 59.2 | 30.6 | 55.6 | 5.6 | 8.3 | 45.9 | 36.5 | 7.1 | 10.6 | .059 | .279 |
| Lack of shelter | 53.3 | 58.3 | 13.9 | 11.1 | 16.7 | 42.4 | 24.7 | 8.2 | 24.7 | .101 | .294 |
| Exposure to frequent and unrelenting fun fire | 53.3 | 38.9 | 47.2 | 2.8 | 11.1 | 50.6 | 37.6 | 5.9 | 5.9 | .127 | .435 |
| Serious physical injury of family/friend due to combat | 52.5 | 69.4 | 27.8 | 0.0 | 2.8 | 76.5 | 22.4 | 0.0 | 1.2 | .249 | .647 |
| Murder, or death due to violence, of other family/friend | 52.5 | 38.9 | 58.3 | 2.8 | 0.0 | 51.8 | 48.2 | 0.0 | 0.0 | .104 | .154 |
| Forced separation from family members | 51.7 | 55.6 | 16.7 | 25.0 | 2.8 | 45.9 | 25.9 | 24.7 | 3.5 | .262 | .692 |
| Witness beatings to head or body | 45.0 | 47.2 | 47.2 | 2.8 | 2.8 | 58.8 | 0.0 | 37.6 | 3.5 | .134 | .309 |
| Beating to the body | 40.0 | 58.3 | 41.7 | 0.0 | 0.0 | 61.2 | 32.9 | 1.2 | 4.7 | .417 | .441 |
| Confiscation or destruction of personal property | 39.2 | 69.4 | 13.9 | 5.6 | 11.1 | 57.6 | 22.4 | 10.6 | 9.4 | .182 | .519 |
| Disappearance or kidnapping of other family/friend | 33.3 | 69.4 | 22.2 | 8.3 | 0.0 | 65.9 | 25.9 | 8.2 | 0.0 | .476 | .912 |
| Enforced isolation from others | 30.8 | 72.2 | 11.1 | 11.1 | 5.6 | 68.2 | 10.6 | 14.1 | 7.1 | .454 | .956 |
| Combat situation (e.g. shelling and grenade attacks) | 25.8 | 58.3 | 41.7 | 0.0 | 0.0 | 81.2 | 18.8 | 0.0 | 0.0 | **.007** | **.012**\* |
| Serious physical injury from combat situation | 25.8 | 52.8 | 44.4 | 2.8 | 0.0 | 45.9 | 49.4 | 1.2 | 3.5 | .362 | .563 |
| Other types of sexual abuse or sexual humiliation | 24.2 | 91.7 | 8.3 | 0.0 | 0.0 | 69.4 | 24.7 | 4.7 | 1.2 | **.007** | **.043**\* |
| Witness killing/murder | 22.5 | 72.2 | 25.0 | 0.0 | 2.8 | 80.0 | 18.8 | 1.2 | 0.0 | .215 | .328 |
| Present while someone searched your home | 20.8 | 77.8 | 0.0 | 19.4 | 2.8 | 80.0 | 16.5 | 2.4 | 1.2 | .451 | .709 |
| Kidnapped | 19.3 | 77.8 | 2.8 | 16.7 | 2.8 | 82.4 | 2.4 | 15.3 | 0.0 | .347 | .480 |
| Someone was forced to betray you, risking death/injury | 19.3 | 77.8 | 19.4 | 2.8 | 0.0 | 82.4 | 0.0 | 16.5 | 1.2 | .347 | .395 |
| Rape | 17.5 | 97.2 | 2.8 | 0.0 | 0.0 | 76.5 | 18.8 | 2.4 | 2.4 | **.004** | **.035**\* |
| Torture (i.e., beating, mutilation, etc) | 16.7 | 80.6 | 13.9 | 2.8 | 2.8 | 84.7 | 12.9 | 2.4 | 0.0 | .352 | .485 |
| Forced labor (like animal or slave) | 14.2 | 86.1 | 11.1 | 0.0 | 2.8 | 85.9 | 10.6 | 2.4 | 1.2 | .593 | .742 |
| Imprisonment | 10.8 | 83.3 | 11.1 | 2.8 | 2.8 | 91.8 | 2.4 | 5.9 | 0.0 | .136 | .053\* |
| Witness torture | 10.8 | 80.6 | 16.7 | 0.0 | 2.8 | 92.9 | 4.7 | 1.2 | 1.2 | **.044** | .088\* |
| Knifing or axing | 7.5 | 86.1 | 13.9 | 0.0 | 0.0 | 95.3 | 4.7 | 0.0 | 0.0 | .081 | .124\* |
| Murder, or death due to violence, of spouse | 5.8 | 100.0 | 0.0 | 0.0 | 0.0 | 91.8 | 8.2 | 0.0 | 0.0 | .083 | .102 |

(*Continued*)

**Table 3.** (Continued)

| | % of Trauma by Full Sample | Males | | | | Females | | | | χ² (sex & 2-level trauma) | χ² (sex & 4-level place) |
|---|---|---|---|---|---|---|---|---|---|---|---|
| | | No Exposure | Home Country | After | Both | No Exposure | Home Country | After | Both | | |
| Disappearance or kidnapping of spouse | 5.8 | 94.4 | 2.8 | 2.8 | 0.0 | 94.1 | 4.7 | 1.2 | 0.0 | .668 | .733 |
| Witness rape or sexual abuse | 5.0 | 100.0 | 0.0 | 0.0 | 0.0 | 92.9 | 3.5 | 3.5 | 0.0 | .120 | .263 |
| Forced to physically harm non-family/friend | 3.3 | 97.2 | 2.8 | 0.0 | 0.0 | 97.6 | 2.4 | 0.0 | 0.0 | .648 | .657 |
| Murder, or death due to violence, of son or daughter | 3.3 | 91.7 | 8.3 | 0.0 | 0.0 | 98.8 | 0.0 | 1.2 | 0.0 | .074 | **.025**[*] |
| Prevented from burying someone | 2.6 | 97.2 | 2.8 | 0.0 | 0.0 | 97.6 | 2.4 | 0.0 | 0.0 | .634 | .657 |
| Forced to physically harm family member, or friend | 2.5 | 91.7 | 8.3 | 0.0 | 0.0 | 98.8 | 1.2 | 0.0 | 0.0 | .074 | .078[*] |
| Forced to destroy someone else's property or possessions | 2.5 | 94.4 | 5.6 | 0.0 | 0.0 | 98.8 | 1.2 | 0.0 | 0.0 | .203 | .211 |
| Forced to betray family/friend risking their death/injury | 2.5 | 97.2 | 2.8 | 0.0 | 0.0 | 97.6 | 2.4 | 0.0 | 0.0 | .648 | .657 |
| Forced to betray non-fam./friend risking their death/injury | 2.5 | 94.4 | 5.6 | 0.0 | 0.0 | 98.8 | 1.2 | 0.0 | 0.0 | .203 | .211 |
| Disappearance or kidnapping of son or daughter | 2.5 | 100.0 | 0.0 | 0.0 | 0.0 | 96.5 | 2.4 | 1.2 | 0.0 | .352 | .521 |
| Forced to find and bury bodies | 1.7 | 97.2 | 2.8 | 0.0 | 0.0 | 98.8 | 0.0 | 1.2 | 0.0 | .500 | .248 |
| Other Trauma Questions | | | | | | | | | | | |
| Have you ever witnessed an armed conflict/confrontation? | 63.3 | 27.8 | 0.0 | 63.9 | 8.3 | 41.2 | 48.2 | 3.5 | 7.1 | .081 | .294 |
| Have you ever been threatened with a firearm? | 50.0 | 30.6 | 47.2 | 11.1 | 11.1 | 58.8 | 32.9 | 2.4 | 5.9 | **.002** | **.011**[*] |
| Have you ever been threatened with any other weapon? | 30.8 | 66.7 | 0.0 | 30.6 | 2.8 | 70.6 | 24.7 | 3.5 | 1.2 | .375 | .563 |
| Have you suffered from natural disaster? | 45.8 | 52.8 | 47.2 | 0.0 | 0.0 | 55.3 | 44.7 | 0.0 | 0.0 | .426 | .843 |

[*] = Fisher's Exact test p-value

significance in the difference between male and female participants; as well as chi-square and Fisher's exact test of difference between male and female participants and the place where trauma occurred (or if it did not occur).

The most endorsed trauma identified was being forced to hide, which was experienced by 75.0% of the sample with the largest percentage of people endorsing that this type of trauma occurred during migration. The places where trauma experiences occurred varied by trauma type, with ten types of traumatic experiences having occurred exclusively in participants' home country (experiencing a war/combat situation, being knifed/axed, experiencing the murder of a spouse, being forced to physically harm both family/friends and non-family/friends, being forced to betray both family/friends and non-family/friends, being forced to destroy the property of others, and experiencing a natural disaster such a floods, droughts, earthquakes).

A war/combat experience only occurred in participants' home country, and men were more likely than women to experience this form of trauma (58.3% vs 18.8%; p = 0.007). Female participants experienced significantly more rape (p = 0.004) and other forms of sexual abuse

(p = 0.007) than men and this difference persisted when we considered place of occurrence (p = 0.035 and p = 0.043 respectively). Experiencing imprisonment was more likely among men than women in their home country, but more likely among women than men during migration. Men were significantly more likely to witness torture than women (p = 0.044), particularly in their home country or in both contexts (before and after leaving their home country). Men were more likely to experience the murder or death due to violence of a son/daughter in their home country, while women were more likely to experience this trauma during their migration journey (p = 0.025). Finally, half of the total sample endorsed being threatened with a firearm. However, men were more likely to be threatened with a firearm than women (p = 0.002), and this difference persisted by place of occurrence (p = 0.011).

"Fig 1. Cumulative Traumas by Sex" illustrates the cumulative number of traumas experienced by male and female participants along with the percentage of each sex in the sample. The percentages on the y-axis refer to percentages of men or women in the sample. Only 1.2% of females (1 participant) and no males endorsed that they had no trauma exposures. Almost 15% of the female participants experienced between 1 and 5 traumas compared to 11% of the male sample. Nearly 27% of female participants experienced between 6 and 10 different traumas compared to 20% of males. However, the percent of males becomes greater than the percent of females experiencing 11 to 15 types of traumas or more than 20 types of traumas.

## Discussion

Our study focuses on the timing and circumstances of trauma exposure of immigrants prior to their arrival to the U.S. and the degree to which exposures were related to sex. Prior work introduces the theory of "trilateral migration trauma" [30] among forced migrant children and families to understand mental health consequences; the framework refers to the potential traumas experienced in three phases: departure, migration, and relocation. The specificity in the timing and location of trauma experiences in our study aligns with the framework in [30] and provides an in-depth perspective of the experiences that may shape decisions to leave the home country versus experiences that happen along the journey to the U.S. Forced migration differs from traditional push-pull factors (e.g. economic opportunity, family reunification) because forced migrants may not always have the option of weighing traditional push-pull factors in their decisions to migrate [30]. Our study is consistent with theory from [30] because many of the immigrants in our sample left their home countries abruptly and under dangerous circumstances because of violence or threats of violence.

Our results reflect that immigrants traverse political instability [24, 31], lack of trust in institutions and impunity [32], and are increasingly vulnerable to being treated as commodities throughout their migration journey [33]. Conflict/war-like situations and experiencing the murders of family are consistent with previous studies about factors driving migration decisions for people seeking safety in the U.S [20, 34]. Our sample shows high levels of experiences of war-like conditions (e.g. 53% were exposed to frequent and unrelenting gun fire and 50% have been personally threatened with firearms) and experienced murders of family members and friends (52% of the sample). As a result of these war-like exposures, it is not unreasonable to think that migrants would seek to migrate away from such violence in the region. For example, prior work [34] demonstrates that past year victimization of self or a family member increases considering family migration to the U.S. by 30% among 49,000 survey respondents across 17 Latin American countries. Our sample is majority female of whom 18.8% are pregnant, and 94% of participants in our study were traveling with children. These two data points shed light on the changing demographics of migrants who are increasingly forced to migrate from their countries in search of safety and the prospect of a better future for their families. As

our data shows, the journey to the U.S. is extremely dangerous, to say the least, for people such as pregnant women and children to undertake, unless they face few other options but to do so.

A majority of both men and women migrants in our study are fleeing and seeking safety rather than migrating for economic reasons (52.2% of respondents identified forced displacement as the main reason for migration). This trend is consistent with data from the United Nations High Commissioner for Refugees (UNHCR) which documents that worldwide more than two in five new asylum applications in 2022 were made by people from Latin America and the Caribbean [35]. Migration decisions often involve a complex set of factors (e.g. violence, lack of economic or educational opportunities, state level persecution or corruption) [30]. Asylum seekers in the U.S. experienced high levels of trauma in their home countries, perpetrated by gangs and state level actors, and also experience the denial of protections from the state when solicited, casting doubt on the safety of individuals who flee their countries in the case of involuntary return or deportation back to Latin American countries [20].

In general men and women experience an equally long list of traumas with two important exceptions. Women experience more sexual violence and rape, and men are exposed to combat-like situations [36] and are more likely to be threatened with firearms. This study sheds light on the extensive availability of firearms to the region [5, 37, 38], causing widespread fear, injury, and death. The rates of exposure to firearm violence in our study are alarmingly high compared to the U.S. adult population. Our participants are more than twice as likely to experience firearm threats compared to all U.S. adults, and twice as likely compared to Black U.S. adults, who experience the highest rates of firearm threats in the U.S. [39].

The ugliness of the migration experience does not discriminate between men and women, however, women disproportionately experience sexual violence. Prior work documents the high levels of sexual trauma perpetrated by gangs particularly in the Northern Triangle countries (Honduras, El Salvador and Guatemala) [20]. While sexual trauma has been documented during migration for many women in previous studies [19, 40], our findings point to significantly higher rates among female migrants (compared to male migrants) in both their home countries and during migration.

Our findings on the rates of rape and sexual victimization are similar to the rates experienced by women in the U.S. [41]. However, prior studies also document that there is a high prevalence of sexual and domestic violence against Latinx immigrant women in the U.S. and a low prevalence of help seeking due to immigration status and other factors [42, 43]. These studies raise a troubling prospect for the future of Latinx women in our sample, many of whom have already experienced high rates of violence, but perhaps whose agency for seeking help may be diminished in the U.S.

The prevalence of traumas experienced in this sample is enormous. Our study demonstrates the intersectional vulnerability of this population, which is shaped by socio-political, economic and gender disparities experienced in the home country and throughout the journey, and the prospect of facing less availability to help seeking and services once they are in the U.S. Our study reveals that individuals completing a migration journey from Latin America are traumatized and would benefit from targeted support and interventions that take the sex of the immigrant into consideration.

## Limitations

There are several limitations to this work. First, the study sample primarily consisted of individuals who came to the U.S. seeking asylum or other forms of protection. As a result, the participants had a high likelihood of experiencing trauma, and potentially reflected in our findings. However, a persistent challenge in conducting research with recent immigrants is the

ability to access populations of recent immigrants to the U.S. It is challenging to access migrants who attempted but were unable to cross into the U.S., or the small proportion of immigrants who remain undetected by border security and other authorities once they are in the U.S. The current study focuses on migrants whom we can locate and access. While it is possible that trauma experiences may differ, studies of these other immigrant populations (e.g., did not cross into the U.S., or remain undetected after crossing into the U.S.) are generally unfeasible.

## Conclusion

This study is among the first to examine exposure to trauma differences by sex and place of occurrence in a population of recently arrived adult immigrants from Latin America. This study draws from a broader set of countries than prior studies. This study has important implications for clinical practice and research as it differentiates timing of traumatic events among male and female adults from Latin America. Interventions focused on reducing symptoms of anxiety, depression and PTSD that are part of a multi-modal model of care could prove beneficial for this population.

## Supporting information

**S1 Checklist.** *PLOS ONE* **clinical studies checklist.**
(DOCX)

## Acknowledgments

The authors would like to thank our community partners at Catholic Charities of the Rio Grande Valley. The authors would also like to thank the participants of this study; without their time and sharing of experiences this research is not possible.

## Author Contributions

**Conceptualization:** Laura X. Vargas, Mary D. Sammel, Therese S. Richmond, Connie M. Ulrich, Zachary D. Giano, Lily Berkowitz, C. Neill Epperson.

**Data curation:** Lily Berkowitz.

**Formal analysis:** Laura X. Vargas, Mary D. Sammel, Therese S. Richmond, Connie M. Ulrich, Zachary D. Giano, Lily Berkowitz, C. Neill Epperson.

**Funding acquisition:** Laura X. Vargas.

**Investigation:** Laura X. Vargas.

**Methodology:** Laura X. Vargas, Therese S. Richmond, Connie M. Ulrich, Zachary D. Giano, Lily Berkowitz, C. Neill Epperson.

**Writing – original draft:** Laura X. Vargas.

**Writing – review & editing:** Laura X. Vargas, Mary D. Sammel, Therese S. Richmond, Connie M. Ulrich, Zachary D. Giano, Lily Berkowitz, C. Neill Epperson.

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
