## [Decision Letter · Decision Letter 0]

8 Aug 2023

PONE-D-23-10833Traumatic experiences and place of occurrence: an analysis of sex differences among a sample of recently arrived immigrant adults from Latin AmericaPLOS ONE

Dear Dr. Vargas,

Thank you for submitting your manuscript to PLOS ONE. After careful consideration, we feel that it has merit but does not fully meet PLOS ONE’s publication criteria as it currently stands. Therefore, we invite you to submit a revised version of the manuscript that addresses the points raised during the review process.

This paper presents an intriguing and original approach to a topic of crucial relevance to public health, particularly at the intersection with various sectoral policies, encompassing a hard-to-reach population. The findings obtained offer an enriching perspective on intersectionality (gender, race, and class) and immigration policies in the United States. Nevertheless, some considerations should be taken into account regarding the reviewers' assessments, aiming to enhance the content and analysis of the study.

Regarding Reviewer 1's evaluation, I agree that the discussion requires a more substantial deepening. While the presented results are information-rich, it's important to delve further into the identified issues. Avoid repeating previously presented results in the earlier section, focusing efforts on a more critical analysis of the implications of these findings. A notable example is the mention of forced displacement as a migration motive, which demands a more thorough analysis to comprehend its ramifications and implications.

Furthermore, I acknowledge Reviewer 1's concern about premature conclusions. The difference in safety perception between men and women during the migration journey needs careful examination. This discrepancy might not solely be attributed to a perception of dangers in the country of origin, as suggested. I suggest revising this section, incorporating theoretical perspectives or additional evidence to support the inferences made by the authors.

Regarding Reviewer 2's suggestions to enhance the organization of the introduction, I concur with the need to reorganize the paragraphs, starting with those beginning at line 85. This will allow for a more logical progression of ideas and a more cohesive introduction.

As for the information about the open-ended question related to migration motives, I agree that it requires greater contextualization. It is advisable to expand this section, providing details about the methodology used in the collection and analysis of immigrants' responses.

Lastly, it's important to clarify that the availability of the study's database is crucial. It's not sufficient to merely state that there are some restrictions and that all relevant data are within the manuscript and its Supporting Information files. I recommend a thorough review of the editorial policy of PLOS ONE.

We look forward to receiving your revised manuscript.

Kind regards,

Ricardo de Mattos Russo Rafael, Ph.D.

Academic Editor

PLOS ONE

2. In the ethics statement in the Methods, you have specified that verbal consent was obtained. Please provide additional details regarding how this consent was documented and witnessed, and state whether this was approved by the IRB

3.  Please provide additional information regarding the considerations made for the immigrants included in this study. For instance, please discuss whether participants were able to opt out of the study and whether individuals who did not participate receive the same treatment offered to participants

Reviewers' comments:

Reviewer's Responses to Questions

**Comments to the Author**

1. Is the manuscript technically sound, and do the data support the conclusions?

Reviewer #1: Yes

Reviewer #2: Yes

2. Has the statistical analysis been performed appropriately and rigorously? 

Reviewer #1: I Don't Know

Reviewer #2: Yes

3. Have the authors made all data underlying the findings in their manuscript fully available?

Reviewer #1: Yes

Reviewer #2: Yes

4. Is the manuscript presented in an intelligible fashion and written in standard English?

Reviewer #1: Yes

Reviewer #2: Yes

5. Review Comments to the Author

Reviewer #1: The authors present an interesting and original manuscript on a topic of relevance to public health, in its interface with other sectoral policies, with a population that is difficult to access.

However, despite the rich results of the study and what they reveal about intersectionality (gender, race, class) and migration policies in the US, it is considered that the discussion was not sufficiently in-depth. In fact, in several occasions, it is perceived in the discussion, a partial repetition of the results already presented without an in-depth analysis of the identified issues or even without any analysis (i.e.: the forced displacement for migrating was just cited on the discussion without any further analysis.).

Other conclusions seem a bit hasty. For example, the difference in perception of safety between men and women during the migration journey might not be due to a perception of dangers in the home country as “normal”.

Reviewer #2: Dear authors,

Congratulations for your study. The theme is of great relevance, as the geopolitical and economic conditions are the central causes for the findings of the study. It is important to identify the consequences of this political action. It was a pleasure to read your work.

I made some comments on your paper, with the intention to help you to align your ideas.

• Perhaps, the introduction would be more organized if the paragraphs starting on line 85 were the first one to appear. Then the paragraphs starting on line 66 can continue the text.

• I do not understand why the information "an open-ended question asking immigrants to briefly explain their reasons for migrating and circumstances that prompted their decision" is mentioned, since nothing else is informed about it.

• It would be interesting if the percentages were homogeneously indicated as numbers, rather than as words or fractions, such as "sixty percent" (line 181) and "three quarters of" (line 182).

6. PLOS authors have the option to publish the peer review history of their article (what does this mean?). If published, this will include your full peer review and any attached files.

Reviewer #1: No

Reviewer #2: No

---

## [Author Response · Author response to Decision Letter 0]

19 Nov 2023

Dear Editor and Reviewers,

Thank you to the Editor for their helpful observations. Overall, we have made modifications to the manuscript based on the Editor’s and Reviewers’ comments. We address the comments of Reviewers below. We believe the suggestions we received have made this manuscript better and we have tried to be responsive to your helpful suggestions. 

- In terms of making the dataset publicly available, we do not have explicit consent from our participants to make our data publicly available. In fact, because of the sensitive nature of this data and the vulnerability of this population, we have told our participants that this data is only available in a de-identified way to members of the research team. We realize that we are in an era of open data, but we also know of the reluctance of certain understudied and extremely important populations to participate in research. Part of our ability to collect this data is the trust that participants place in us. We value their trust greatly and it is vital to respect our commitment to the trust placed in us. However, as requested by the journal, we provide a non-author point of contact where data requests may be sent.

We thank Reviewer #1 for their helpful comments and insights. Based on the feedback we received we have:

o Edited the discussion section to reflect a more in-depth analysis of our results, by placing it within existing theory and evidence in the field. 

o We appreciate the observation that we were repeating some of our results, and we have eliminated repeating of findings the discussion and instead include a more in-depth analysis of what our results may mean for the current state of the literature. 

o We have revised some of the conclusions that are considered premature and provided an overall better context for our discussion section. 

Thank you to Reviewer #2 for their helpful suggestions and comments. In response to them we have made the following changes:

o We moved the paragraphs in the introduction according to the Reviewer’s suggestions. 

o We have clarified the responses for the question on the reason for migration. We clarify that actually, rather than an open-ended answer (we described it incorrectly in our previous draft), the reason for migration is a multiple choice response and we clarify what the possible answers to this question are in the methods section of the manuscript.

o We modified the language to reflect percentages as numbers

We thank you again for considering this manuscript for peer-review in your Journal. Please let us know if there is anything else you may need. 

Sincerely,

Laura Vargas, PhD, LMSW, MPA (corresponding author)

---

## [Decision Letter · Decision Letter 1]

31 Jan 2024

PONE-D-23-10833R1Traumatic experiences and place of occurrence: an analysis of sex differences among a sample of recently arrived immigrant adults from Latin AmericaPLOS ONE

Dear Dr. Vargas,

Thank you for submitting your manuscript to PLOS ONE. After careful consideration, we feel that it has merit but does not fully meet PLOS ONE’s publication criteria as it currently stands. Therefore, we invite you to submit a revised version of the manuscript that addresses the points raised during the review process.

**I would like to express my sincere appreciation for your consideration of the suggestions submitted during the last round of evaluation. I kindly request that you once again take into account the comments provided by the reviewer.**

We look forward to receiving your revised manuscript.

Kind regards,

Ricardo de Mattos Russo Rafael, Ph.D.

Academic Editor

PLOS ONE

Journal Requirements:

Reviewers' comments:

Reviewer's Responses to Questions

**Comments to the Author**

1. If the authors have adequately addressed your comments raised in a previous round of review and you feel that this manuscript is now acceptable for publication, you may indicate that here to bypass the “Comments to the Author” section, enter your conflict of interest statement in the “Confidential to Editor” section, and submit your "Accept" recommendation.

Reviewer #1: All comments have been addressed

2. Is the manuscript technically sound, and do the data support the conclusions?

Reviewer #1: Yes

3. Has the statistical analysis been performed appropriately and rigorously? 

Reviewer #1: I Don't Know

4. Have the authors made all data underlying the findings in their manuscript fully available?

Reviewer #1: Yes

5. Is the manuscript presented in an intelligible fashion and written in standard English?

Reviewer #1: No

6. Review Comments to the Author

Reviewer #1: The authors made significant changes to the article to allow its publication. However, a review regarding some aspects is still necessary. Some doubts persist, especially concerning the rationale behind the choices of the data that the authors propose to analyze, despite of others. For example, we have the percentage of pregnant women or men whose wives were pregnant, which seemed non-negligible (15.6%), but is not mentioned in the article as a significant and noteworthy data. Overall, it seems that the richness of the findings deserves a more extensive discussion, or that the choices should be explicitly stated. Other minor considerations: 1) p.13 l. 252 - it is suggested to avoid expressions like "past year" and instead use the specific year in question; 2) p.13, l. 254 to 258: The second sentence does not justify the first statement, as the authors attempt to do; 3) references 32 to 35 appear for the first time in the text after references 36 to 38.

7. PLOS authors have the option to publish the peer review history of their article (what does this mean?). If published, this will include your full peer review and any attached files.

Reviewer #1: No

---

## [Author Response · Author response to Decision Letter 1]

7 Mar 2024

We thank Reviewer #1 once again for their helpful comments and insights. To address the reviewer’s comments we made the following changes to our submission:

- We thank the reviewer for their comment on explicitly stating the rationale for our analysis. We add a sentence at the end of our Introduction section doing so, which we hope will be helpful. 

- We thank the reviewers for highlighting the importance of the proportion of our sample who are pregnant or traveling with a pregnant partner. We make reference to this portion of our sample in more detail in the result page 6 lines 179-181, and again in our discussion of results in page 13 lines 261-266.

- We thank the reviewer for their comment regarding the richness of our discussion. WE have tried to incorporate some of the suggestions of reviewers in the previous round of comments and in this round of comments in the hopes that we can improve this manuscript. We also try to balance the need for our manuscript to be clear, concise, and precise in its content to meet the journal’s very reasonable requirements.

- We clarify in the measures section of our manuscript what is meant by the “previous year” page 5, lines 140-141. Because the survey questions is meant to ask about employment in the 12 months prior to the date of interviews, we do not refer to a specific year of employment because our interviews were conducted over the course of several weeks/months the year in reference may be different. In addition, the way we phrase the question is “how many months did you work in the previous 12 months?” – an explanation that we include in our manuscript.

- We included two clarifying sentences in page 13, lines 256-260 about how war-like conditions in the Latin American region may be linked to migrants’ decisions to leave their home countries.

- We have made the proper adjustments for references to appear in sequential order.

---

## [Decision Letter · Decision Letter 2]

3 Apr 2024

Traumatic experiences and place of occurrence: an analysis of sex differences among a sample of recently arrived immigrant adults from Latin America

PONE-D-23-10833R2

Dear Dr. Vargas,

We’re pleased to inform you that your manuscript has been judged scientifically suitable for publication and will be formally accepted for publication once it meets all outstanding technical requirements.

Kind regards,

Ricardo de Mattos Russo Rafael, Ph.D.

Academic Editor

PLOS ONE

Reviewers' comments:

Reviewer's Responses to Questions

**Comments to the Author**

1. If the authors have adequately addressed your comments raised in a previous round of review and you feel that this manuscript is now acceptable for publication, you may indicate that here to bypass the “Comments to the Author” section, enter your conflict of interest statement in the “Confidential to Editor” section, and submit your "Accept" recommendation.

Reviewer #1: All comments have been addressed

2. Is the manuscript technically sound, and do the data support the conclusions?

Reviewer #1: Yes

3. Has the statistical analysis been performed appropriately and rigorously? 

Reviewer #1: I Don't Know

4. Have the authors made all data underlying the findings in their manuscript fully available?

Reviewer #1: Yes

5. Is the manuscript presented in an intelligible fashion and written in standard English?

Reviewer #1: Yes

6. Review Comments to the Author

Reviewer #1: Good quality paper, subject of major interest, highly recommended publication. All comments have been addressed.

7. PLOS authors have the option to publish the peer review history of their article (what does this mean?). If published, this will include your full peer review and any attached files.

Reviewer #1: **Yes: **Clarissa Terenzi Seixas

---

## [Editor Report · Acceptance letter]

24 May 2024

PONE-D-23-10833R2 

PLOS ONE

Dear Dr. Vargas, 

I'm pleased to inform you that your manuscript has been deemed suitable for publication in PLOS ONE. Congratulations! Your manuscript is now being handed over to our production team.

Kind regards, 

on behalf of

Dr. Ricardo de Mattos Russo Rafael 

Academic Editor

PLOS ONE